# Pseudotyped virus infection of multiplexed ACE2 libraries reveals SARS-CoV-2 variant shifts in receptor usage

**Nidhi Shukla**, **Sarah M. Roelle**, **John C. Snell**, **Olivia DelSignore**, **Anna M. Bruchez**, **Kenneth A. Matreyek** *

Department of Pathology, Case Western Reserve University School of Medicine, Cleveland, Ohio, United States of America

* kenneth.matreyek@case.edu

**Data Availability Statement:** All relevant data are within the manuscript, its Supporting Information files, and online repositories. Raw sequencing files can be found at NCBI Geo accession number

## Abstract

Pairwise compatibility between virus and host proteins can dictate the outcome of infection. During transmission, both inter- and intraspecies variabilities in receptor protein sequences can impact cell susceptibility. Many viruses possess mutable viral entry proteins and the patterns of host compatibility can shift as the viral protein sequence changes. This combinatorial sequence space between virus and host is poorly understood, as traditional experimental approaches lack the throughput to simultaneously test all possible combinations of protein sequences. Here, we created a pseudotyped virus infection assay where a multiplexed target-cell library of host receptor variants can be assayed simultaneously using a DNA barcode sequencing readout. We applied this assay to test a panel of 30 ACE2 orthologs or human sequence mutants for infectability by the original SARS-CoV-2 spike protein or the Alpha, Beta, Gamma, Delta, and Omicron BA1 variant spikes. We compared these results to an analysis of the structural shifts that occurred for each variant spike's interface with human ACE2. Mutated residues were directly involved in the largest shifts, although there were also widespread indirect effects altering interface structure. The N501Y substitution in spike conferred a large structural shift for interaction with ACE2, which was partially recreated by indirect distal substitutions in Delta, which does not harbor N501Y. The structural shifts from N501Y greatly influenced the set of animal orthologs the variant spike was capable of interacting with. Out of the thirteen non-human orthologs, ten exhibited unique patterns of variant-specific compatibility, demonstrating that spike sequence changes during human transmission can toggle ACE2 compatibility and potential susceptibility of other animal species, and cumulatively increase overall compatibilities as new variants emerge. These experiments provide a blueprint for similar large-scale assessments of protein compatibility during entry by diverse viruses. This dataset demonstrates the complex compatibility relationships that occur between variable interacting host and virus proteins.

GSE255644. The code and processed data files capable of recreating the entire analyses can be found at https://github.com/MatreyekLab/SARS2-CoV-2_ACE2_variant_combinations.

**Funding:** This research was supported by National Institutes of Health (NIH) grants GM142886 (KAM), AI156907 (KAM), AI178151 (KAM), R21AI161275 (AMB), and R21AI169561 (AMB). The funders had no role in study design, data collection and analysis, decision to publish, or preparation of the manuscript.

**Competing interests:** The authors have declared that no competing interests exist.

## Author summary

The proteins encoded by a viral pathogen must be compatible with the proteins encoded by the host organism to allow the virus to enter and take over their cells. As the SARS-CoV-2 pandemic progressed, various SARS-CoV-2 variants of concern emerged, each altering the sequence of their spike proteins. We developed a new high-throughput infection assay which allowed us to simultaneously test dozens of sequence variants of ACE2 for their ability to allow viral particles coated with the spike proteins from the SARS-CoV-2 variants of concern to enter the cell. We found that the SARS-CoV-2 variant spike proteins only slightly changed in the way they interacted with human ACE2 but exhibited more pronounced changes in their ability to interact with ACE2 versions from other species. The SARS-CoV-2 spike variants generated during the pandemic were collectively compatible with the majority of animal ACE2 sequence versions we tested. We speculate that similar molecular interplays may occur during cross-species transmission, where repeated transmission and escape from immune responses allows the virus to simultaneously explore new molecular compatibilities with different receptor protein versions, thus allowing spread to new species.

## Introduction

The ability of a virus to enter a cell is necessary but not sufficient for infection of the host. The viral entry protein must be adequately compatible and capable of interacting with the host cell receptor to allow viral entry to occur. Some viruses like influenza have host receptors with limited variability, such as the α2-3-linked sialic acid bound by avian influenza, as compared to the α2-6-linked sialic acid bound by human influenza viruses [1]. Other viruses use proteinaceous receptors, which generally exhibit more variability due to the inherent amino acid sequence permutability of proteins. The host receptor protein sequence can be variable, either through polymorphisms within a species, or through divergent evolution between species. In some cases, even single amino acid differences in the protein sequence can change the overall likelihood of infection [2]. Viruses are highly adaptable by regularly generating sequence variants, and select for those with improved fitness over the original sequence, particularly in new environments. Such complex pairwise interplays are particularly relevant when considering potential routes of cross-species transmission.

The interaction between the SARS-CoV-2 spike receptor binding domain (RBD) and its host ACE2 receptor is influenced by sequence variants in both protein partners. There is generally little sequence variation within ACE2 in the human population, but rationally engineered mutants of the human ACE2 have revealed functionally important residues within the interaction interface [3,4]. ACE2 sequence is highly variable between species and is positively selected in bats [5]. Studies that tested sets of animal ACE2 orthologs revealed the broad range of ortholog compatibility exhibited by the original Wuhan / WA1 isolate RBD [6–10], but differences in experimental details can confound comparisons of ortholog compatibility between studies.

The extended transmission of SARS-CoV-2 that occurred during the pandemic generated waves of viral variants that passed through the population, each cresting at different times in different geographical locations. Few studies have systematically examined how each variant spike RBD differentially relies on the human ACE2 protein surface, or those from the aforementioned panels of animal ACE2 orthologs. Such a comprehensive pairwise assessment of SARS-CoV-2 spike RBD variants and ACE2 orthologs can reveal which animals were

susceptible to past variants, and reveal general patterns in shifting compatibility which may inform compatibility assessments with future spike variants.

Here, we performed an analysis of 13 variants of human ACE2 and 13 animal ACE2 orthologs and tested them against 6 different SARS-CoV-2 variant spike proteins in a cell-based viral entry model system. By tagging each transgenic ACE2 sequence with a predetermined DNA barcode, we were able to test all ACE2 orthologs simultaneously within the same mixed pool of cells. We calculated the relative infection rates promoted by each ACE2 sequence by performing high throughput DNA sequencing of susceptible cells that became infected with pseudotyped virus, and dividing those barcode frequencies by the frequencies of the barcodes present irrespective of infection. We confirmed spike variant-specific shifts in human ACE2 surface interaction, and furthermore revealed patterns in ACE2 animal-ortholog compatibilities exhibited by the SARS-CoV-2 spike variants.

## Results

### Library-based SARS-CoV-2 entry assays with receptor variants

To enable our studies, we improved upon our arrayed fluorescence-based pseudovirus infection scheme [3,11] by adapting it to a multiplexable high-throughput sequencing readout. We appended each of our transgenic expression constructs with a unique pair of nucleotide identifiers, together forming a nucleotide barcode [12]. By giving each construct a unique predetermined barcode, we could identify the relative frequencies of a particular transgenic cDNA within a mixed population by sequencing only the barcode region with high-throughput Illumina sequencing (**Fig 1A**). These barcoded plasmids, each encoding the same Bxb1 attB recombination site but a different transgenic cDNA cargo, can be mixed together and transfected *en masse* into specialized HEK 293T "Landing Pad" cells [3,12,13]. These cells encode a singular genomic Bxb1 attP site, so that the Bxb1 integrase enzyme can only genomically integrate a single plasmid from the pool of transfected plasmid molecules. The plasmids are promoterless, so only the singular transgenic cDNA integrated into the genomically embedded attP site in the cell is expressed from a pre-engineered tet-inducible promoter. This method establishes a strict genotype-phenotype link at the single cell level.

We appended unique barcode combinations to a set of 30 plasmids encoding two subsets of samples: ACE2 orthologs from a range of animals, including the reference sequence of human ACE2, and various mutants of human ACE2, with a subset known to affect SARS-CoV or SARS-CoV-2 entry [3] (**S1 Table**). The ACE2 orthologs were largely from horseshoe bats of the genus *Rhinolophus*, originally chosen as a panel of diverse ACE2 sequences that represented potential hosts for distantly related sarbecoviruses. In aggregate, this diverse set of samples helped demonstrate the strengths and limitations of this assay format for studying virus entry. These ACE2 sequences were also previously studied with SARS-CoV-2 D614G spike using a traditional pseudotyped virus infection format, and were also easily obtained, making them convenient and effective benchmarks for the multiplex infection assay.

Equimolar amounts of each plasmid were mixed to create a barcoded ACE2 library, which were recombined into the landing pad cells (**Fig 1B**). Following selection, we possessed a library of cells each expressing a single WT or mutant cDNA from the barcoded ACE2 plasmid pool. These cells were mixed with pseudotyped virus harboring a lentiviral vector encoding the green fluorescent protein mNeonGreen and the hygromycin resistance gene hygromycin phosphotransferase, coated with the original Wuhan / WA1 SARS-CoV-2 spike with the early D614G mutation, which we will simply refer to as "D614G" hereafter (**Fig 1C**; see methods for details about immunoblot confirmation of spike incorporation).

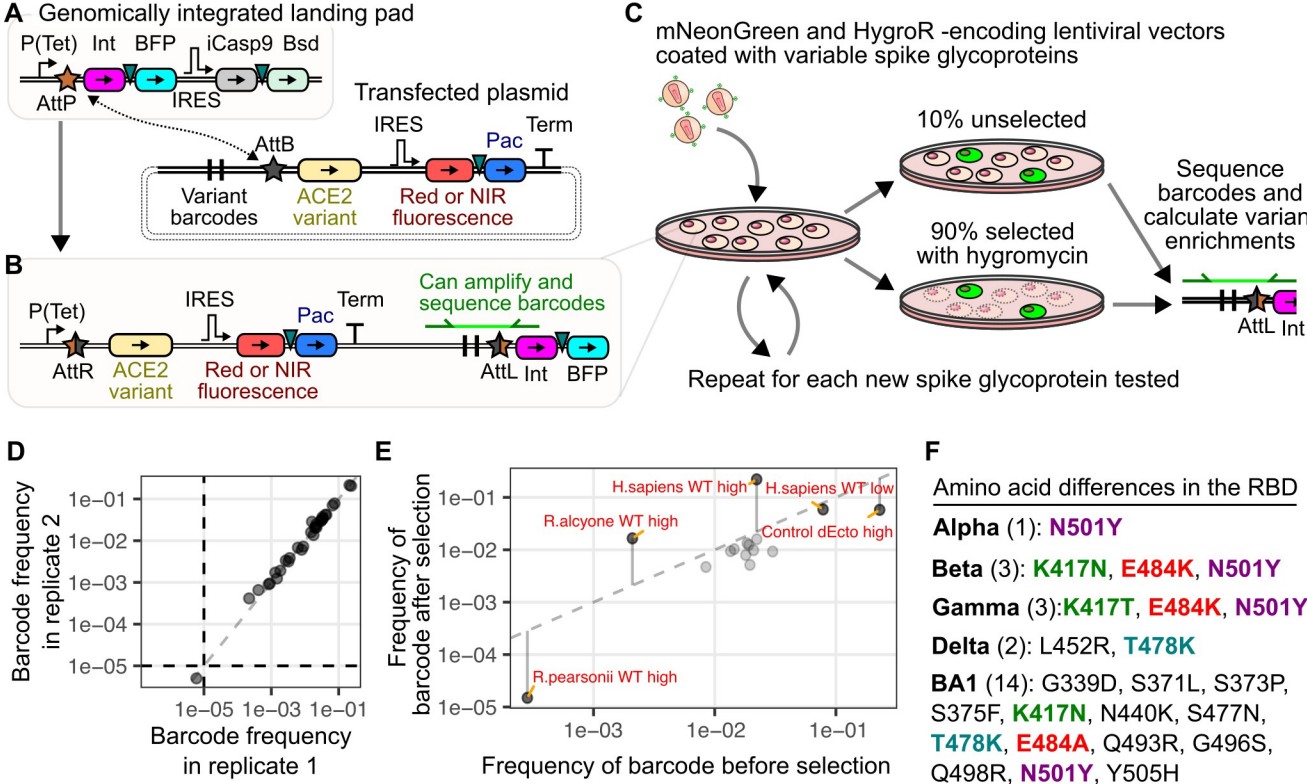

**Fig 1. Infection of barcoded ACE2 expressing cells with pseudotyped virus.** A, B) DNA construct schematics for generating the barcoded receptor cell library. Schematics of the genomic landing pad and transfected plasmid molecules prior to recombination are shown in panel A, while the result of plasmid integration into the genomic landing pad upon recombination is shown in panel B. Triangles represent 2A-like peptides used for co-translational protein separation. C) Workflow diagram of the infection, cell collection, and barcode DNA sequencing scheme. D) Representative scatter plot showing barcode frequency correlations in replicate unselected cell cultures. E) Representative scatter plot of the changes in barcode frequencies that occurred in an infected cell population before and after selection with hygromycin, to enrich for infected cells. Datapoint labels "high" and "low" refer to ACE2 abundance level, while "dEcto" lacks the ectodomain of ACE2 that is necessary for interaction with viral spike protein and is included as a negative control. F) List of amino acid differences observed in the RBDs of the spike proteins tested in our experiment. Substitutions at identical RBD positions are bolded and colored by position. Abbreviations: Tet, tetracycline-inducible promoter; Int, Bxb1 DNA integrase; BFP, blue fluorescent protein; IRES, internal ribosome entry site; iCasp9, inducible caspase 9; Bsd, blasticidin S deaminase; Pac, puromycin N-acetyltransferase; Term, Simian Virus 40 terminator sequence; attP and attB, Bxb1 sites prior to recombination; attR and attL, Bxb1 sites following recombination; HygroR, hygromycin resistance gene.

Roughly 10% of the culture was collected and stored prior to selection, while the remaining 90% of the culture was selected with hygromycin to enrich for infected cells. Genomic DNA was extracted from both samples, amplified for the barcoded regions (**Fig 1C**), and sequenced with an Illumina NextSeq high throughput short read sequencer to count each barcode. The ratio of the frequency of each barcode, and by extension each ACE2 cDNA, in the selected and unselected samples yielded enrichment scores used to characterize the viral entry-mediating function of each ACE2 cDNA within the library.

We observed that replicate extractions of unselected cells yielded correlated results (**Fig 1D**; Pearson's $r^2$ = 0.99). When we compared the frequencies of the ACE2 cDNAs before and after selection, the control samples exhibited the expected effects (**Fig 1E**). Theoretically, the absence of any selection (**Fig 1D**) or randomly acting selection would yield approximate identical frequencies in the before and after selection samples (**Fig 1E, dotted line**). We observed that samples expected to enhance the rate of infection, such as human ACE2 or the *R.alcyone* bat ACE2 ortholog previously shown to be highly susceptible to SARS-CoV-2 infection [11], increased in frequency following selection with hygromycin. In contrast, samples expected to

have minimal infection, such as human ACE2 missing its entire ectodomain sequence (dEcto), or the *R.pearsonii* bat ACE2 ortholog previously shown to not be susceptible to SARS-CoV-2 infection [11,14], had reduced frequency following selection. We also observed intermediate effects, such as cells with human ACE2 at 50-fold reduced levels relative to the high-abundance samples, which exhibit low-level pseudovirus entry [3]. These cells were neither enriched nor depleted relative to the other samples in the mixture following selection (**Fig 1E**).

Having established the ability of our assay to simultaneously assess infectivity of multiple ACE2 orthologs and human ACE2 mutants with pseudotyped lentivirus coated with spike proteins with the Wuhan/WA1 SARS-CoV-2 RBD (D614G), we expanded our experiment to include spike proteins from the original variants of concern to understand how spike-ACE2 interactions have shifted over the initial course of the pandemic. We exposed the cell library with a panel of SARS-CoV-2 spike variant pseudoviruses. This included the original spike protein supplemented with the D614G change (labeled as D614G), or the Alpha, Beta, Gamma, Delta, and Omicron BA1 variants (**Fig 1F**).

While all samples were experimentally assayed in aggregate, there were two general groups of ACE2 samples present in the library which were generally analyzed separately for simplicity of interpretation (**Fig 2A**). One group represented human ACE2 mutants expressed at low abundance levels roughly 50-fold less than our high abundance level cells (**Fig 2A, bottom**), as our previous work showed that relatively subtle missense changes of the human ACE2 sequence had little effect on infection by SARS-CoV or SARS-CoV-2 spike pseudotyped

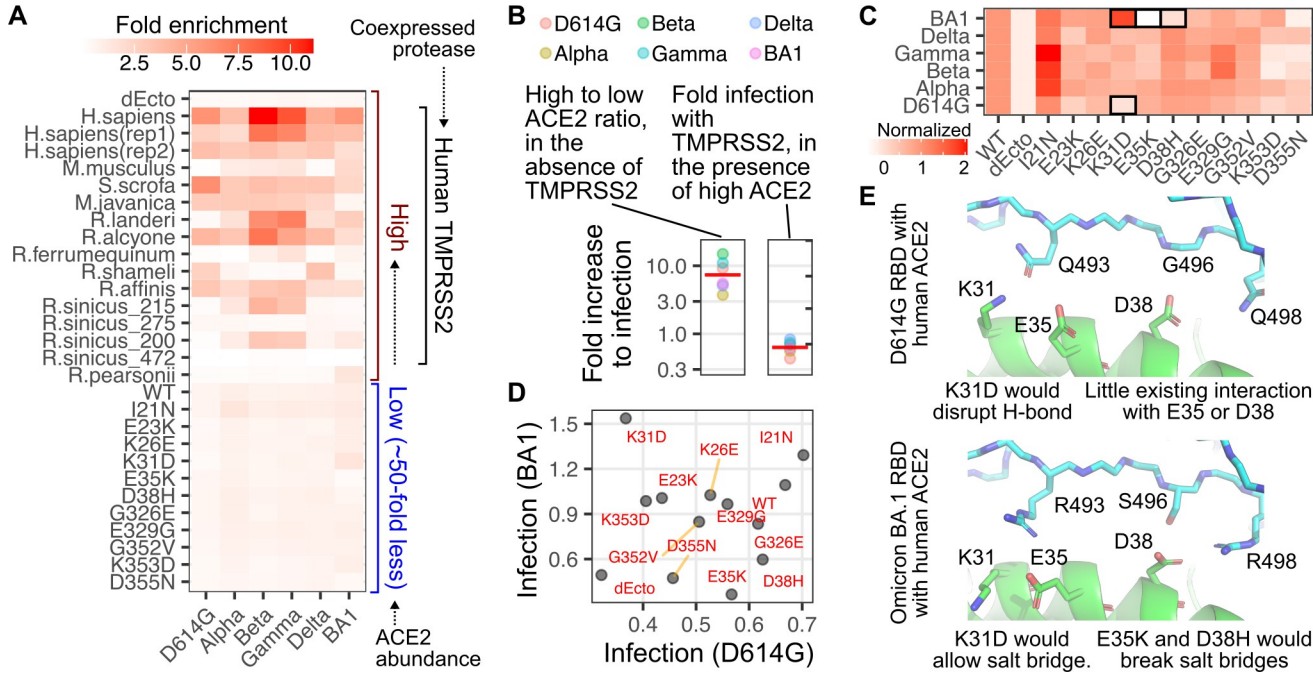

**Fig 2. ACE2 enrichment scores following SARS-CoV-2 spike pseudotyped infection.** A) Heatmap of geometric mean fold enrichment values across replicates for all samples in the library, without rescaling. The bottom-half consists of human ACE2 mutants. B, left) The ratio of infectivity for human ACE2 expressed at high abundance divided by infectivity at low abundance. The red line indicates the geometric mean value across variants. B, right) The ratio of infectivity for cells with high human ACE2 abundance in the presence of co-expressed TMPRSS2 divided by infectivity in the absence of TMPRSS2. The red line indicates the geometric mean value across variants. C) Infection values of human ACE2 mutants expressed at low abundances, rescaled to WT set at 1 and the ectodomain-deleted ACE2, dEcto, set at 0. Mutants of notable phenotypes discussed in the text are highlighted with black boxes. D) Scatter plot comparing unscaled, fold enrichment values of human ACE2 mutant cells at low abundances infected with D614G spike or BA1 spike pseudotyped viruses. E) Structural representations of the human ACE2 helix spanning Lys31, Glu35, and Asp38, in complex with D614G RBD (pdb: 6m17) or Omicron BA1 RBD (pdb: 7t9k).

viruses when expressed at high abundance levels [3]. This is likely since the avidity-based interactions that occur during viral entry are functionally unaffected by slight mutational decreases to already high affinity interactions with human ACE2 when the number of ACE2 proteins on the cell surface are saturating. We observed an average 10-fold enhancement to infection across all of the SARS-CoV-2 variants when infecting high versus low human ACE2 abundance cells (**Fig 2B, left**), consistent with our previous results with more traditional infectivity readouts [3].

ACE2 sequence is highly diverse across animals, and there are many animal orthologs known to have little to no affinity for SARS-CoV or SARS-CoV-2 spike, such as the murine ortholog from *M.musculus*. Thus, the other group of samples consisted of various animal orthologs of ACE2 expressed at high abundance (**Fig 2A, top**), to reveal potential enhancements to SARS-CoV-2 spike -mediated infection even when the proteins were only partially compatible [11]. Each of these constructs also expressed human TMPRSS2, although we saw no enhancement to human ACE2 -mediated infection when TMPRSS2 was present (**Fig 2B, right**). This is likely because the endosomal route of infection was sufficient to enable full infectivity when ACE2 was overexpressed in these engineered cultured HEK 293T cells.

We first assessed the impacts of single amino acid changes to human ACE2 on variant infectivities. To improve interpretation, we normalized the enrichment values to that of WT human ACE2, which was set to 1. These values were moderately correlated with previously measured infectivities from a traditional infection assay (**S1 Fig**). To more readily reveal patterns in the impacts of single amino acid mutants, we further scaled the data so that the negative control cells (dEcto) possessed a value of 0 (**Fig 2C**).

Most SARS-CoV-2 variants infected the cells harboring various human ACE2 mutants similarly, with a few exceptions. We previously observed that SARS-CoV-2 D614G spike pseudotyped viruses were sensitive to a D355N change in ACE2 [3], and similar effects were seen for all variants, with Omicron BA1 the most sensitive within the set (**Fig 2C**). D614G spike was disrupted by the K353D change in ACE2, and a N501Y substitution could compensate for this [3]. We indeed found that infection with viruses pseudotyped with Alpha spike, harboring only the N501Y substitution in the RBD, was largely unaffected by the K353D mutant (**Fig 2C**). The underlying reasons appear complex and did not purely segregate with all spikes harboring N501Y, as Delta and Omicron were similarly unaffected, while Beta and Gamma were as sensitive as D614G spike.

Omicron BA1 spike, by nature of its large set of substitutions at the interaction interface, yielded a pattern of ACE2 mutational sensitivity most dissimilar to the others (**Fig 2C**). SARS-CoV-2 D614G spike pseudotyped virus poorly entered cells expressing the K31D mutant of human ACE2, while infection with SARS-CoV-2 Omicron BA1 spike pseudotyped virus was enhanced (**Fig 2D**). This is likely due to formation of a salt bridge between Arg493 on the BA1 spike and the Asp side-chain from the ACE2 K31D mutant, which would be disrupted and potentially repulsive with a Lys residue (**Fig 2E**).

Infection with the Omicron BA1 spike pseudotyped virus was disrupted by the ACE2 E35K and D38H mutants. Looking at experimentally determined structures between the BA1 spike RBD and human ACE2, this seems likely due to a salt bridge that forms between Asp38 in ACE2 and Arg498 of the spike protein (**Fig 2E, bottom**). This salt bridge would not be present in any other SARS-CoV-2 variants, as they all encode Gln498 at the comparable position. The ACE2 E35K substitution seems to clash with the Arg493 side chain in Omicron BA1, which is Gln493 in the other tested variant spikes. Importantly, the fold enhancement of infection for low-abundance human ACE2 cells was quite low, and our current data was insufficiently powered to statistically analyze differences between the samples (**S1C Fig**), which may indicate a limitation of using multiplex assays to study samples of relatively minor effect compared to

other samples in the pool. Altogether, these results showed that our infection data can reveal interplays in molecular compatibilities that dictate ACE2 receptor usage by different SARS-CoV-2 variants, although broader patterns involving multiple points will still be more statistically robust than interpretations with specific data samples, particular for those with low magnitudes of effect in the multiplex setting.

## Structural analysis of altered human ACE2 engagement by variant spikes

Dozens of human ACE2 and SARS-CoV-2 variant RBD interaction structures have been independently resolved through X-ray crystallography or cryo-electron microscopy. We wished to incorporate this structural data into our analysis, as it could reveal key points of interaction that can be modulated through amino acid substitutions. We performed a meta-analysis of 25 co-structures between human ACE2 and these variant spike RBDs (S2 Table), calculating the minimum atomic distances between each residue in the spike RBD and human ACE2, for four structures with Wuhan / WA1 RBD, three structures with Alpha RBD, five structures with Beta RBD, four structures with Gamma RBD, four structures with Delta RBD, and five structures with Omicron BA1 RBD (S2A Fig). We took the mean values between all distance calculations for a given ACE2-RBD residue pair to yield an average ensemble distance for each variant (S2B Fig). Overall, the ACE2-RBD pairs with the highest coefficients of variation across the ensemble datasets were pairs at known ACE2 interaction residues paired with RBD residues mutated in at least one spike variant (S2C Fig).

We focused our analyses on potentially key interacting pairs between the spike RBDs and ACE2. We first identified pairs of ACE2 and spike RBD residues that were within 10 angstroms in all structures. This distance would not only capture residue pairs in direct non-covalent bonds, such as hydrogen bonds, salt bridges, or hydrophobic effect, but also identify more indirect relationships that may occur through changes to protein tertiary structures. We then performed pairwise comparisons between each of the computed distance matrices, and identified pairs of residues that were outliers for the otherwise normal distribution (S3 Fig), suggesting bona-fide variant-specific shifts in atomic distance between these residue pairs.

To further hone the list to the most critical interactions, we focused on the pairs found outside of the normal distribution in at least 20% of all spike variant comparisons, yielding a set of 25 ACE2:RBD residue interaction pairs. We bolstered this list with 21 interaction pairs that were observed to be within 3.4 angstrom distances in at least one of the ensemble distance matrices, thus including relatively strong and direct noncovalent interactions that may only undergo subtle variant-specific shifts not originally captured with the aforementioned criteria. The final list had 38 interaction pairs between 18 ACE2 positions and 18 RBD positions. Plotting these positions on the structural model confirmed that these residues were at the interaction interface (Fig 3A). Despite the dataset consisting of a mixture of 8 structures determined by X-ray crystallography and 17 structures determined by cryo-electron microscopy, the ACE2 interaction pair distances for each RBD variant were generally grouped together regardless of acquisition method (S4 Fig), suggesting that protein sequence was bigger driver for the observed distance changes rather than potentially artifactual effects conferred by structural acquisition method.

We created a force-directed network diagram to visualize these interactions (Fig 3B). ACE2 residues Asp38, Gln42, and Lys353, along with D614G RBD residues Asn501 and Gln498, were the most interconnected residues, each exhibiting 4 or more connections to residues from the partner protein, with at least one connection exhibiting a distance closer than 4 angstroms (Fig 3B). We next visualized how each connection changed with different variant RBDs (Fig 3C).

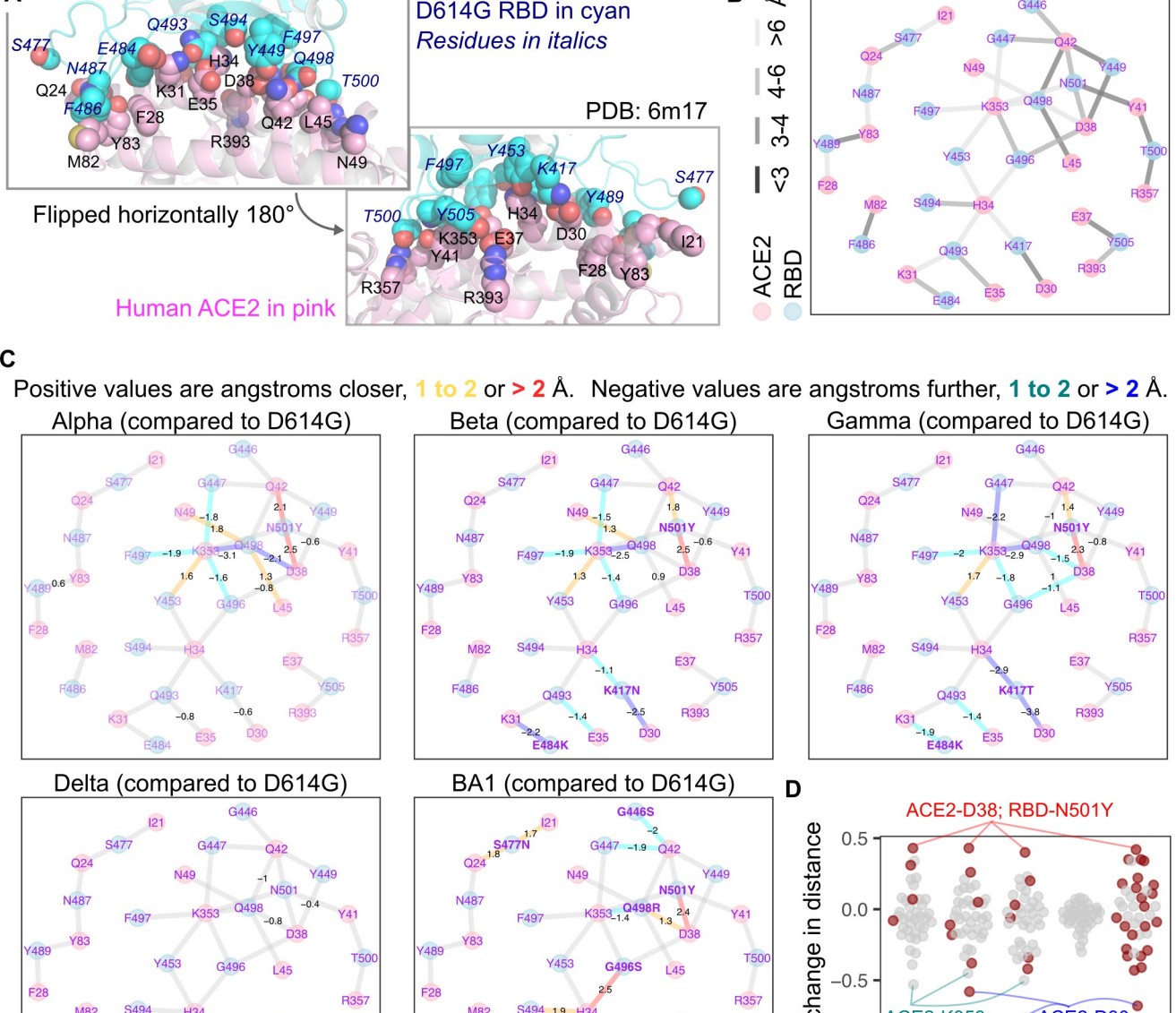

**Fig 3. Structural shifts observed between SARS-CoV-2 spike and human ACE2.** A) Structural representation of the ACE2 and D614G RBD interaction interface, with interaction side chains identified and considered in our analysis shown as spheres. The RBD is shown in cyan, while ACE2 is shown in pink. Italicized blue labels indicate D614G RBD residues, while black labels are ACE2 residues. B) Force-directed network diagram showing nodes and edges connecting the ACE2 and RBD interaction residues. The lightness of the gray coloring of the edges indicate approximate atomic distances in angstroms. C) The same network diagram shown in panel B, but with values indicating significant shifts in atomic distance that occurred for each variant RBD structure as compared to D614G. Values are in angstroms, with positive values having moved closer, and negative values having moved further away. Edges are also colored according to this shift. Mutated RBD residues are shown in bold. D) Fraction of initial D614G atomic distance for each interaction pair, with closer shifts positive and negative shifts further. Red dots indicate interaction pairs where the RBD residue was mutated in that variant. Notable interaction pairs are labeled.

The N501Y mutation in Alpha caused a drastic change in the network of interactions between the RBD and ACE2. The new Tyr501 residue more closely interacted with ACE2 residues Asp38 and Gln42. The Gln498 residue in the RBD drastically shifted away from both

ACE2 residues Asp38 and Lys353, while moving closer to Asn49 and Leu45. ACE2 Lys353 also shifted away from RBD residues Phe497, Gly496, and Gly447, while moving closer to Tyr453. Very similar shifts occurred with the Beta and Gamma variants, which also harbor the N501Y mutation along with two additional changes in the RBD. Delta encodes the original Asn501 residue and did not exhibit the same shifts, although there were similar increases to distances between ACE2 Asp38, ACE2 Gln42, RBD Gln498, and RBD Tyr449, that were observed with Alpha, Beta, and Gamma. Omicron BA1, which encodes N501Y, also resulted in the now Tyr501 residue moving closer to ACE2 Asp38, although the other shifts were not seen, likely due to the presence of nearby mutations Q498R, G446S, and G496S in this variant.

Beta and Gamma differ from Alpha by the presence of two additional mutations; E484K and K417N/T. The E484K mutation increased the distance to ACE2 Lys31. The K417N and K417T mutations increased the distance to ACE2 Asp30 and His34. Omicron BA1 encodes E484A and K417N mutations, and demonstrated the same shifts away from Lys31, Asp30, and His34. Delta, despite not encoding mutations at either Glu484 or Lys417, exhibited similar albeit milder shifts away from these same residues, likely through indirect effects caused by its more distal L452R and T478K mutations. Regardless, the convergent shifts occurring in these major SARS-CoV-2 variant lineages through distinct sets of mutations suggest this phenomenon to be a key step during SARS-CoV-2 transmission and adaptation in humans during the pandemic.

Omicron BA1, with its increased collection of mutations, yielded shifts to the RBD-ACE2 interface not observed with any of the previous variants. The G496S mutation brought the RBD Ser496 residue closer to ACE2 His34, which also moved closer to RBD residue Ser494. The Y505H substitution increased the distance between His505 and ACE2 Glu37 and Arg393. The S477N mutation brought RBD Asn477 closer to Ile21 and Gln24. The G446S mutation, in the context of the nearby N501Y change, appeared to increase the distance between ACE2 Gln42 and RBD residues Ser446 and Gly447.

Expectedly, the largest relative shifts in pairwise interaction distances involved residues mutated on the variant RBD (**Fig 3D**). N501Y consistently reduced RBD Tyr501 proximity to ACE2 Asp38 by roughly 2.5 angstroms. In contrast, the K417N or K417T changes in RBD increased distance to ACE2 Asp30, by 2.4, 3.7, and 3.1 angstroms respectively for Beta, Gamma, and BA1 (**Fig 3D**). The only consistent exception was an increased distance observed between Lys353 on ACE2 and Gln498 on the Alpha, Beta, and Gamma variant RBDs, which was an indirect effect mediated by RBD N501Y. BA1 did not exhibit this effect, but that was likely because of the presence of Arg at position 498 instead of Gln. Delta, which did not encode any mutants at direct interaction sites, exhibited comparatively little change to interaction distances compared with D614G.

## Correlating structural shifts with infectivity

The observed shifts in structure provide context for the ACE2 mutant infection data. The favoring of BA1 spike for mutant Asp31 over Lys31 in ACE2 happens in a context where the BA1 Ala484 and Arg493 mutant sites had already become more distant from ACE2 Lys31 (**Fig 4A**). The favoring of WT ACE2 Asp38 over the His38 mutant, or WT Glu35 over the Lys35 mutant, coincides with BA1 spike residues having moved closer to these positions relative to D614G. Not all shifts closer were functionally impactful, as BA1 spike Asn477 moved closer to ACE2 Ile21, yet mutation of this residue to Asn had little functional effect (**Fig 4A**).

We finally compared how SARS-CoV-2 variant sequence differences, structural changes, and perturbation to infection by ACE2 mutation, each related to each other (**Fig 4B–4D**). BA1 exhibited the largest sequence deviation from the rest of the SARS-CoV-2 variants (**Fig 4B**),

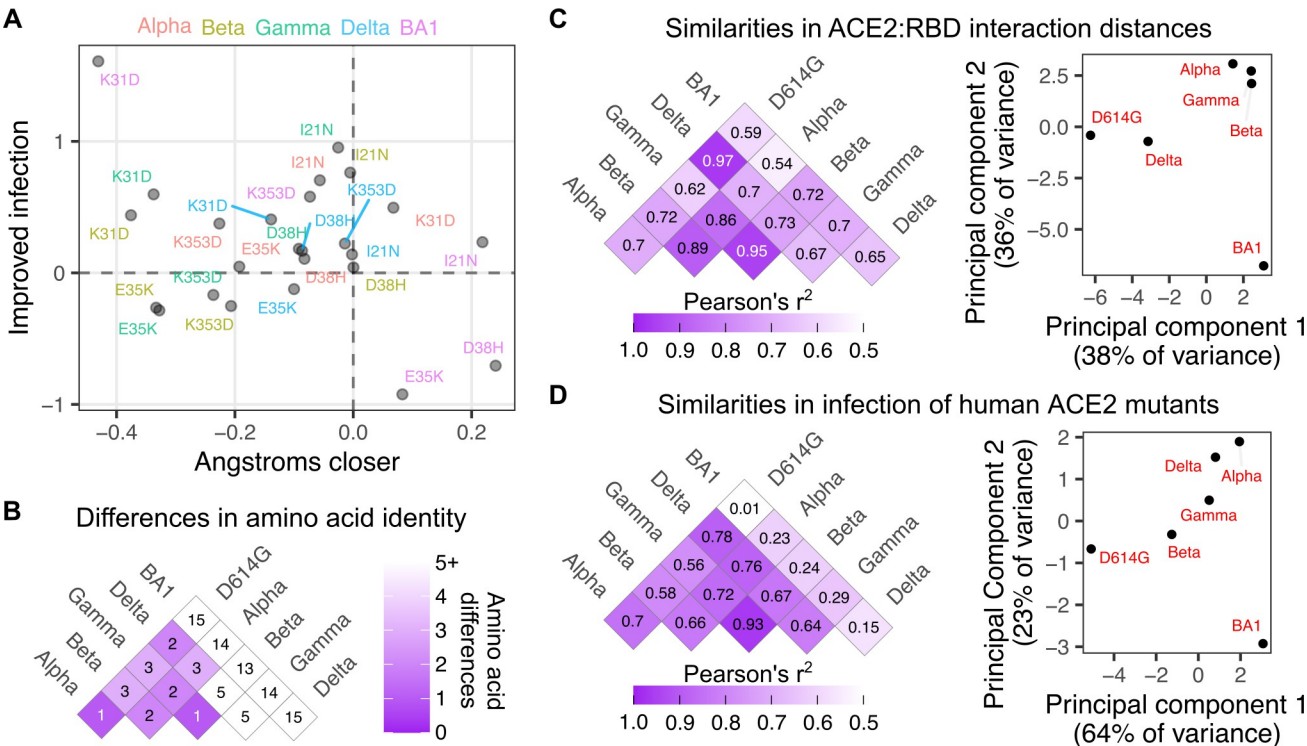

**Fig 4. Comparative shifts in sequence, structure, and human ACE2 mutational tolerance.** A) Scatter plot comparing how key ACE2 residues shifted in distance to the closest RBD interaction residue in each variant, and how log10-normalized infection changed relative to WT human ACE2 infection when the ACE2 residue was mutated to the indicated amino acid. B) Amino acid sequence identity matrix comparison between SARS-CoV-2 variant RBDs. C) Pairwise Pearson's $r^2$ values for ensemble structural similarity between variant RBD (left), and scatter plot of the first two principal components of the structural differences between variant RBDs (right). D) Same as the analyses in panel C, but for similarities and differences in how each variant RBD infected human ACE2 mutant cells.

and correspondingly exhibited the greatest structural shifts and altered sensitivity to ACE2 mutant infection (**Fig 4D**). Alpha, Beta, and Gamma, largely driven by a handful of sequence differences centered around the shared N501Y change, exhibited generally similar structural shifts (**Fig 4C**) and related, albeit not identical, sensitivities to the ACE2 mutant panel (**Fig 4D**). Delta exhibited a unique set of amino acid changes within the RBD not shared by the other variants (L452R and T478K), but these changes still partially recapitulated the structural shifts that occurred with the other variants, and altered sensitivities to the ACE2 mutant panel appeared intermediate between the Beta/Gamma group and Alpha (**Fig 4C and 4D**). Particularly when expanded to future variants, similar analyses will likely be helpful in understanding how SARS-CoV-2 variant sequence changes can converge in various patterns of structural shifts and functional reliances on key ACE2 interface residues during infection.

## SARS-CoV-2 variant infectivity with ACE2 orthologs

The factors governing sequence compatibility between SARS-CoV-2 RBDs and diverse animal ACE2 orthologs are comparatively more complex as the overall effect on infection will be dependent on a combination of simultaneous amino acid changes. ACE2 sequence is variable between animals, with ~ 75 or more changes within the first 400 amino acids of the protein (comprising the entirety of the interaction domain) between distantly related species, and many orthologs across different horseshoe bats, all from the same genus *Rhinolophus*, differing

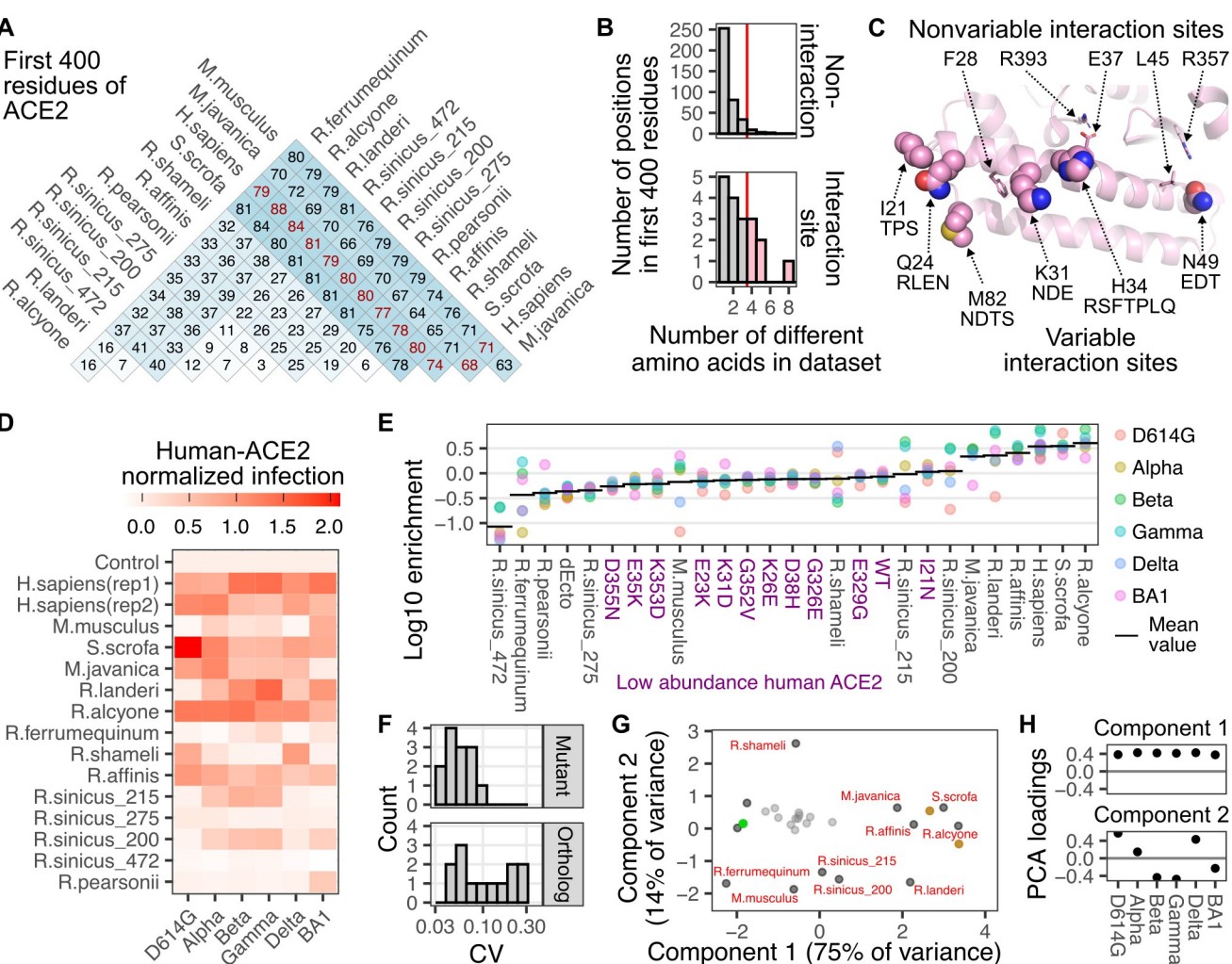

**Fig 5. Sequence differences in ACE2 orthologs underlie infectivity with SARS-CoV-2 variant spikes.** A) Amino acid sequence identity matrix of the first 400 residues of the various animal ACE2 orthologs that were tested in our assay. B) Histograms showing the number of different amino acids observed per ACE2 position, separated by whether they were considered variable SARS-CoV-2 variant interaction residues (bottom) or not (top). The red line represents the threshold value used to identify the six highly variable interaction site residues displayed as spheres in the following panel. C) Structural representation showing the ACE2 residues structurally involved in SARS-CoV-2 interaction which were also highly variable in our ortholog set (spheres), or those that were invariant (sticks). Other amino acids seen in animal orthologs at that position are listed below the human residue. D) Heatmap of efficiencies to infection for each animal ortholog, scaled to the mean of the two human ACE2 samples set to a value of 1, and the ectodomain-deleted control set to 0. E) Grouped scatter plot showing log-10 enrichment values for each spike variant with each ACE2 ortholog. The horizontal bar denotes the geometric mean value across variants for that ortholog. Purple labels denote human ACE2 mutants expressed at low abundance levels. F) Histograms of coefficient of variation values for data points in panel D, separated by human ACE2 mutants at low abundance levels, or ACE2 orthologs at high abundance levels. G) Scatter plot of the first two principal components stemming from spike variant infectivities for each ortholog. The small gray points denote human ACE2 mutants expressed at low abundance. Values corresponding to the two constructs (differing only in their fluorescent reporter, and designated rep1 and rep2 in panel D) encoding human ACE2 at high abundance levels are shown as brown points. Values for the ectodomain-deleted ACE2 expressing control cells is shown as a green point. H) Variant-specific infectivity loadings of the first two principal components shown in Panel G.

from each other by 30 amino acids (**Fig 5A**). Even alleles within the same *Rhinolophus sinicus* species differed by as many as 12 amino acids within our tested set (**Fig 5A**).

The ACE2 residues that exhibited the highest sequence variability in our dataset of animal orthologs only partially overlapped in position with the human ACE2 residues we previously identified as key for interacting with the SARS-CoV-2 spike variant RBDs (**Fig 5B**). The positions corresponding to Ile21, Gln24, Lys31, His34, Asn49, and Met82 on human ACE2 were both identified in our structural meta-analysis and exhibited high sequence variability in our

set of ACE2 orthologs (**Fig 5C**). As multiple contact residues simultaneously differ when comparing one ACE2 animal ortholog to another, we were unable to clearly ascribe functional importance to any specific residue change with this dataset.

We observed a wide range of infectivities of the various animal orthologs to the set of SARS-CoV-2 variants (**Fig 5D and 5E**). Some orthologs were consistently highly infected by the panel of RBD variants; alongside human ACE2, including orthologs from pigs (*S.scrofa*), and horseshoe bats *R.alcyone* and *R.affinis*. There were also orthologs that were consistently poorly infectable. This included two alleles from the chinese horseshoe bats (*R.sinicus_275 and R.sinicus_472*), which exhibit differences from the other tested alleles at key interaction sites (**S5 Fig**). Protein expression from these cDNAs were observed by Western blotting in previous publications [11,15], suggesting that the lack of infection here was not simply due to a total lack of protein made. Importantly, the magnitude of infection enhancement observed between the samples within this dataset yielded effect sizes that clearly distinguished cells based on their infectability (**S1D Fig**).

There were also orthologs that showed SARS-CoV-2 variant-specific compatibilities. Orthologs from pangolins (*M.javanica*) and the *R.landeri* horseshoe bat were generally highly infectable, with a single variant being an exception. *R.pearsonii* exhibited nearly universally poor infection, with the exception of the BA1 spike. These variant-specific orthologs yielded high coefficients of variation values, which were clearly separable from the infections of human ACE2, or those with human-like susceptibility (**Fig 5F**). These different phenotypic groups were visible following a principal component analysis, where plotting the values for the two principal components separated the orthologs with differential variant-specific infectivity in two-dimensional space (**Fig 5G**). The first two principal components corresponded to overall infectivity to all pseudotyped viruses, followed by separation of variants based on Beta/Gamma/BA1-common signatures of infectivity, likely driven by phenotypic changes from substitutions at Asn501, Lys417, and Glu484, amongst other residues (**Fig 5H**).

To further parse these patterns in variant-specific infectivities, we assessed which variant spikes exhibited ortholog compatibility similar to each other using Pearson's correlation coefficients (**Fig 6A**). The sample-specific infectivities for Beta and Gamma corresponded almost perfectly (Pearson's $r^2$ of 0.97; **Fig 6B, left**), which is expected as these two RBD sequences are nearly identical. The next most similar pair was D614G and Delta (Pearson's $r^2$ of 0.83; **Fig 6B, right**). In contrast, there was substantial deviation for many of the other pairwise correlations (**Fig 6A**).

These highly correlated RBD pairs exhibited relatively few differences in their RBD sequences, so we assessed how the number of amino acid differences corresponded with ortholog compatibility across the full set of RBD variants (**Fig 6C**). BA1 exhibits 13 or more differences from all of the other variants, and expectedly, was the least correlated (**Fig 6C, pink filled points**). This included the least correlated pair in the dataset, which was between D614G and BA1 (Pearson's $r^2$ of 0.18; **Fig 6D, left**). Notably, the full set of 15 amino acid differences between D614G and BA1 were seemingly not required to drive this loss of correlation. For example, D614G exhibited a similarly weak functional correlation with Gamma (Pearson's $r^2$ of 0.2; **Fig 6D, right**), despite the two RBDs only differing by 3 amino acids. Thus, there was substantial shifting of animal ACE2 ortholog compatibility possible through relatively minimal change from the original Wuhan / WA1 RBD sequence, which already occurred with the first series of SARS-CoV-2 variant spikes that emerged in 2020.

We observed variant-specific favoring or disfavoring of particular animal orthologs. These pairwise comparisons revealed ACE2 ortholog sequences that differed in their variant infectivities. By using the high abundance human ACE2 and the ectodomain-deleted human ACE2 sample (both without TMPRSS2 co-expressed) to fit a line, we approximated a hypothetical

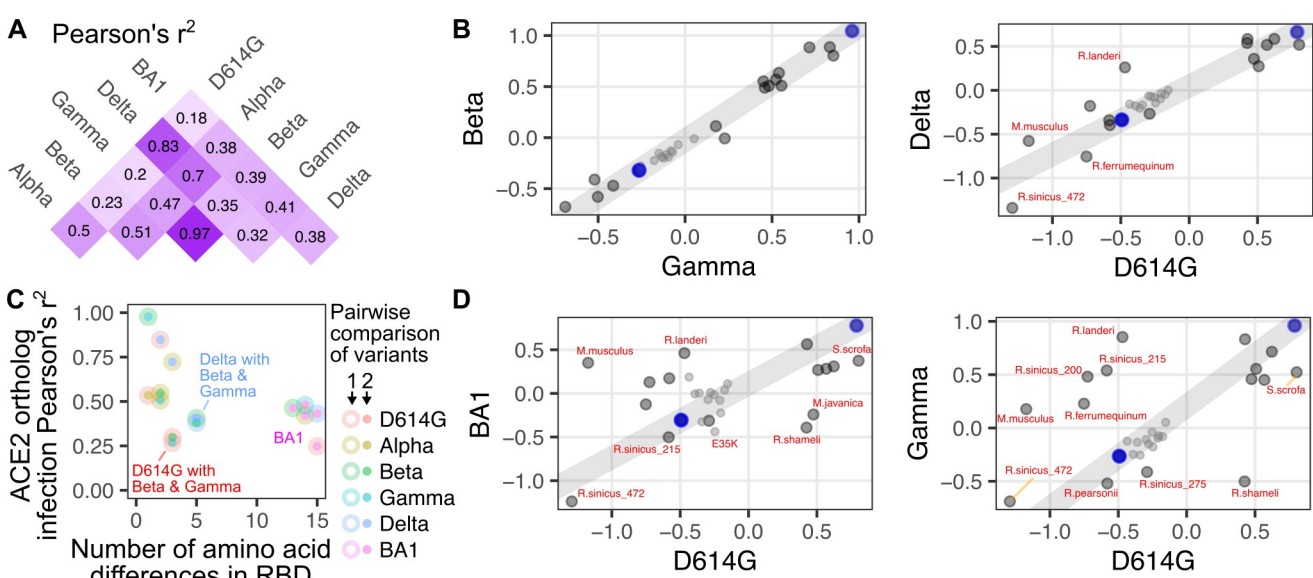

**Fig 6. Comprehensive pairwise comparisons of ortholog-specific compatibility for RBD variants.** A) Squared Pearson's correlation coefficients for all pairwise comparisons of spike variant infectivities. B) Scatter plots of two highly correlated infection values comparing Gamma with Beta variant spike (left), and D614G with Delta variant spike (right). C) Scatter plot comparing the number of amino acid differences between each spike variant pair, and the Pearson's $r^2$ value corresponding to that pair. D) Scatter plots of two poorly correlated infection values comparing D614G with BA1 variant spike (left), and D614G with Gamma variant spike (right). The blue dots in panels B and D represent the high human ACE2 and ectodomain-deleted ACE2 samples used to create the gray line upon which the residuals of each point were calculated. The small gray points in panel B and D denote human ACE2 mutants expressed at low abundance.

direct correspondence between the relative infectivities for any two variants being compared (**Fig 6B and 6D, gray line**). The two replicate samples of human ACE2 with TMPRSS2 co-expressed had slightly reduced infectivity but relatively little deviation from this line. Similarly, all of the human ACE2 mutants at low abundance had substantially reduced infectivity but were all generally positioned overlapping with the line (**Fig 6B and 6D, small points**), with few exceptions. To focus on ACE2 ortholog sequences that defied this relationship established by human ACE2, we identified orthologs that deviated the most from this line by looking at those with the highest residual values. We created matrices of these residuals to identify patterns in variant-specific ortholog usage.

As there were three or fewer amino acid differences between the non-Omicron RBDs, we assessed whether we could ascribe key functional impacts to particular amino acid changes in the viral spike protein on ortholog compatibility. The simplest interpretations were made by comparing Alpha with D614G, since these RBDs only differ by the N501Y substitution (**Fig 7A**). Infection of murine ACE2 (*M.musculus*) is possible with the N501Y substitution in spike [16–18], and we indeed observed increased infection with the Alpha, as well as the other N501Y encoding RBDs from Beta, Gamma, and BA1 spikes (**Fig 7B**). Previous studies have shown that Asn501 is largely incompatible with rodent ACE2 His353, as murine ACE2 with a H353K mutation can restore infection with virus encoding Asn501 RBD [19]. Human ACE2 with a K353H substitution reduces infectivity with Asn501 RBD virus [20], and the N501Y change circumvents these incompatibilities. This is consistent with the indirect distancing we observed between Lys353 and Gln498 that occurred upon N501Y mutation (**Fig 3**).

We observed more complex derivations from simple N501Y-driven patterns. The Alpha RBD exhibited enhanced compatibility with *R.sinicus_200* and *R.sinicus_215* (**Fig 7C and 7D**), but *R.sinicus_215* seemingly lost compatibility with BA1, likely due to one or more of the 14

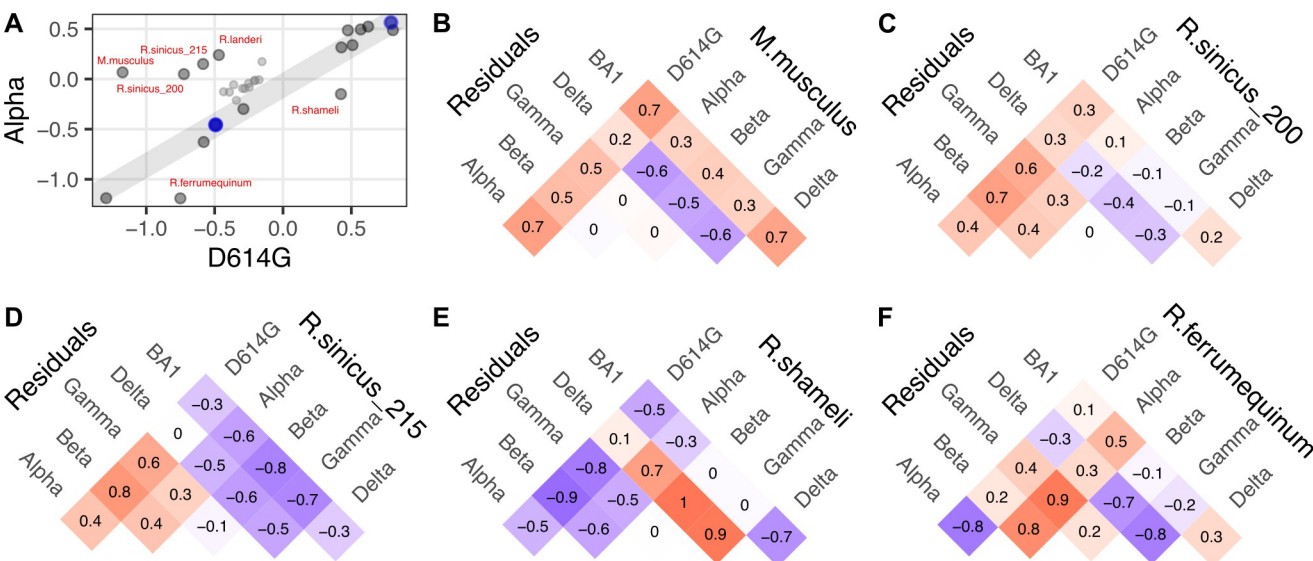

**Fig 7. Putative N501Y-driven patterns in ACE2 ortholog usage.** A) Scatter plot comparing the infection values from D614G with Alpha variant spike, which only differ in RBD sequence at position 501. B, C, D, E, F) Residual values, quantitating deviation from the line of infectivity enhancement provided by human ACE2, for mouse ACE2 (B), as well as four horseshoe bats, including two orthologs of chinese horseshoe bats (C, D), Shameli's horseshoe bat (E), and the greater horseshoe bat, *R.ferrumequinum* (F). Positive values, colored in red, denote pairs where infection was more efficient for the variant on the left as compared to the variant on the right. Negative values, colored in purple, denote pairs where infection was less efficient for the variant on the left as compared to the variant on the right.

other changes present in this RBD. Despite the initial effect with N501Y, both orthologs exhibited improved infection with the Beta or Gamma RBDs, suggesting that this effect was not necessarily conferred by the Glu484 and Lys417 changes.

The same analysis indicated that N501Y reduced compatibility for interacting with the ACE2 ortholog from *R.shameli*, as every variant except for Delta exhibited drastically reduced infection (**Fig 7E**). By itself, N501Y also reduced compatibility with the ACE2 ortholog from *R.ferrumequinum*, as Alpha exhibited vastly reduced infectivity with this ACE2 ortholog as compared to D614G. This sensitivity was furthermore modulated by substitutions at other RBD positions such as Lys417 and Glu484, as Beta, Gamma, and BA1 variants did not exhibit this same sensitivity (**Fig 7F**). Altogether, these results support the importance of the N501Y change in the SARS-CoV-2 variants as a major driver of shifting ACE2 ortholog compatibility.

There were instances where the large number of new substitutions in the Omicron BA1 RBD completely switched ortholog compatibility as compared to the previous SARS-CoV-2 variants. This was clearest when comparing the ortholog infectivities between the Beta and BA1 RBDs, as they share identical N501Y and K417N changes (**Fig 8A**). As compared to every other variant tested, omicron BA1 was poor at infecting cells expressing the ACE2 from pangolin (*M.javanica*; **Fig 8B**), while it was adept at infecting cells expressing ACE2 from Pearsonii's horseshoe bat (*R.pearsonii*; **Fig 8C**). Due to the large number of differences that are present with each of these orthologs and the next most related sequences, it is currently impossible to pinpoint which amino acid features account for these effects.

Despite the small sample sets, there were variant-specific differences in how each virus infected cells with the various ACE2 animal orthologs as compared to the human ACE2 mutants. As expected, Omicron BA1 had the least correlation to any other variant in either context likely driven by its sequence divergence (**Fig 9A**). In contrast, the biggest discrepancies occurred between the Beta / Gamma subset with D614G and to a lesser extent Delta, as they exhibited much more similarity when infecting the human ACE2 mutant panel as compared

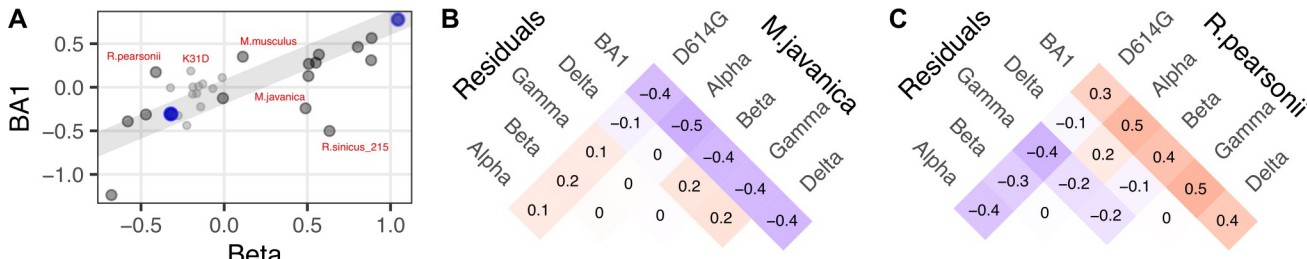

**Fig 8. Unique ACE2 ortholog usages observed with BA1.** A) Scatter plot comparing the infection values from Beta with BA1 variant spike proteins. B, C) Residual values, quantifying deviation from the line of infectivity enhancement associated with human ACE2, for pangolin (B) and Pearsonii's horseshoe bat (C).

to the animal orthologs (**Fig 9A**). Thus, the initial set of three changes at RBD positions 501, 484, and 417 had minor effects on altering sensitivity to single amino-acid substitutions of human ACE2, yet seemingly had substantial effects on interactions with more diverse animal orthologs that exhibited multiple simultaneous differences from human ACE2. Since the human ACE2 mutant panel was tested at low abundance levels to accentuate their impacts on infection [3], these discrepancies would likely be even more pronounced if tested at the high abundance levels used when testing the animal orthologs, as saturating protein abundance appears to mask the effects of slight reductions in binding affinity [3].

While no singular variant simultaneously exhibited high compatibility with all tested orthologs, the shifting of infection compatibilities across variants suggests that the cumulative range of ortholog compatibilities increases as new variants emerge. To consider this, we first identified ACE2 orthologs that conferred a similar amount of susceptibility to a particular variant spike as human ACE2 (**Fig 9B**). We considered these variants as "compatible" with that spike protein (**Fig 9B, green**). While there were intermediate infection pairs below this level but above background (**Fig 9B, gray**), we considered these pairs incompatible for the purposes of our subsequent analysis.

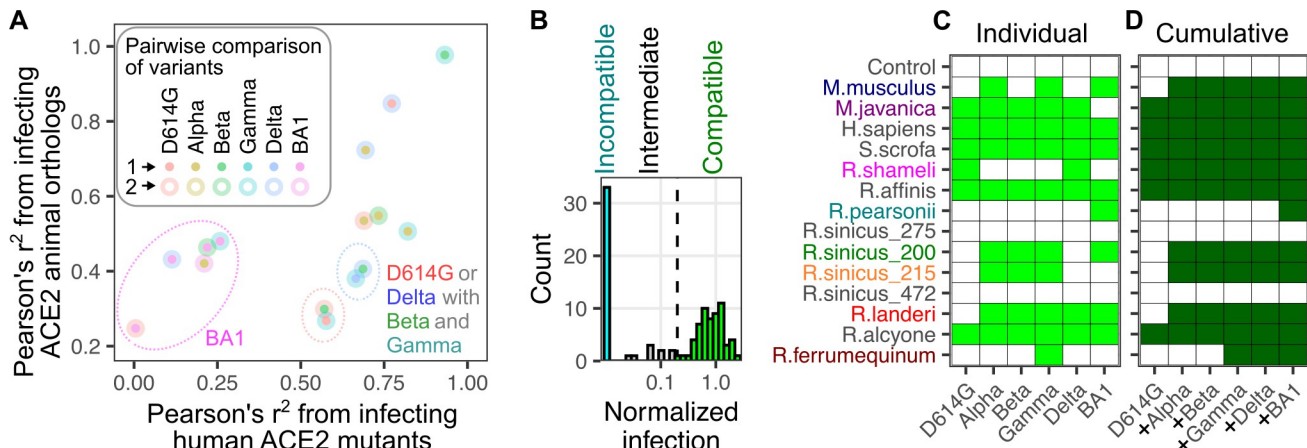

**Fig 9. Cumulative species compatibility with RBD sequence variation.** A) Scatter plot comparing how each pair of RBDs correlate when comparing their infection of human ACE2 mutants (x-axis) or infection of animal ACE2 orthologs (y-axis). Dotted ovals indicate nearby points that can be grouped by a common virus within the comparison. B) Histogram of normalized infection values for the animal ACE2 orthologs, colored based on those considered compatible, intermediate, or incompatible. C) Matrix of variant-specific animal ortholog compatibilities. Green squares denote compatible combinations, while white squares are incompatible. D) Matrix of cumulative animal ortholog compatibilities when variant compatibilities are added to previous ortholog compatibilities, going in order of the Greek alphabet. Dark green squares denote cumulative variant compatibility, going from left to right, for each ACE2 ortholog, while white squares are incompatible with current and previously listed variants.

The original D614G RBD was compatible with 6 of the 14 orthologs tested (**Fig 9C**). We then considered the patterns of infection compatibility for each SARS-CoV-2 variant, following the order of the Greek alphabet, as it roughly marked chronological order of variant emergence (**Fig 9C**). The N501Y change in Alpha conferred four new compatibilities. Gamma provided a novel compatibility to *R.ferrumequinum*. Despite the drastic shift in ortholog infection compatibilities exhibited by Omicron BA1, it only provided one ortholog compatibility with *R.pearsonii* that was not already possessed by a previous SARS-CoV-2 variant. Thus, the original spike RBD was compatible with slightly less than half of the ACE2 orthologs we tested, but the additional compatibilities provided by the variant spikes meant that, in aggregate, nearly all of the tested orthologs were infectable by at least one variant RBD through the emergence of Omicron BA1 (**Fig 9D**).

## Discussion

The multiplexed target cell infection assay described in this work is a scalable cultured-cell model of viral entry that provides quantitative measurements of relative infectivities between samples. There are unique advantages this assay format possesses over traditional arrayed approaches. Having the full panel of receptors present within the same mixed pool of cells allows every sample to be exposed to the same inoculum of pseudovirus, minimizing inter-sample experimental variability. Highly infectable cells are unlikely to naturally exist as a monolayer, and their interspersal amongst poorly infectable cells within a multiplex assay approach may capture aspects of viral infection not normally encountered with assays with a clonal cell line. Our multiplexed assay format better recapitulates this situation, as arrayed approaches require different samples to be kept separate to maintain their identity.

This assay enables the collection of large-scale datasets. Unlike arrayed approaches, where infection with a new pseudotyped virus requires individually generating, counting, and handling dozens of separate target cells kept in different wells, the multiplexed target cell approach only requires generation and infection of a singular, large well containing a pool of all samples mixed together. This streamlining allows inclusion of additional samples or conditions that would be unfeasible through traditional approaches, allowing for richer datasets. While we vetted this assay format with a comparatively modest set of ~ 30 different types of target cells, the same infection, sequencing, and data analysis workflows can be used to compare hundreds or thousands of samples simultaneously. Only the library preparation and barcoding steps would need to be changed, as a ligation-based barcoding and subassembly scheme, such as that possible with PacBio long read sequencing [21], is necessary to link protein sequence variants and DNA barcodes at scale.

This assay is generalizable, and different viral entry interactions can be studied by swapping out the barcoded transgenic constructs and the viral entry glycoproteins on the pseudotyped viruses. Our existing library of ACE2 target cells can be screened with distantly related beta coronavirus RBDs, such as those from BtKY72 and Khosta-2 [11], or other known or putative ACE2-dependent alpha or beta coronaviruses, such as NL63 [22] and NeoCoV [23], to characterize ACE2-RBD interactions that are more diverse than those occurring with the SARS-CoV-2 variant spikes. Presumably, any enveloped virus where the fusion machinery can be reconstituted to coat a lentiviral vector can be tested with this assay framework. For example, NPC1 orthologs from various animals could be tested to learn about the species compatibility of various filovirus envelope glycoproteins [24]. In cases where a particular clade of viruses uses multiple receptor interactions, the entire set of putative receptors can be tested within the same library.

For SARS-CoV-2 spike-mediated entry, the plethora of data generated during the pandemic allowed us to extrapolate upon the infection assay results. The multiplexed SARS-CoV-2 spike

infection results we generated correlated well with previous data collected through more traditional assays (S1 Fig), helping to vet the assay format. The multitude of independently determined co-structures between human ACE2 and each SARS-CoV-2 variant spike RBD provided experimental replication, where the ensemble structure can reveal reproducible patterns of effects, separable from those caused by experimental variability. The availability of these structures allowed us to make comparisons examining how structure corresponds to function.

There was remarkable convergence in the structural shifts that occurred with the Alpha, Beta, and Gamma RBDs relative to the original sequence. The N501Y substitution created major shifts in the networks of interactions involving ACE2 Asp38, ACE2 Lys353, and RBD Gln498. Even Alpha, with the sole N501Y change to its RBD, showed slight distancing between ACE2 Glu35 and RBD Gln493, along with ACE2 Asp30 and RBD Lys417. This effect was accentuated in the Beta and Gamma RBDs through direct K417N/T substitutions. Delta, despite having two unrelated changes in L452R and T478K, showed similar shifts as those produced more directly by N501Y and K417N/T. Delta even exhibited distancing between ACE2 Lys31 and Spike Glu484, similar albeit to reduced magnitude as effects of the more direct E484K substitution present in Beta and Gamma.

This commonality in structural outcome through independent mutational routes suggests that these structural shifts conferred a fitness advantage. The variant spikes exhibited relatively similar sensitivities to the human ACE2 mutants tested within our panel, so any adaptations to human ACE2 binding were likely minor tweaks rather than gross shifts in compatibility. This is also reflected in more traditional binding affinity measurements made between purified SARS-CoV-2 spike RBDs and human ACE2, wherein the published dissociation constant ($K_D$) values for SARS-CoV-2 spike variant RBDs interacting with human ACE2 consistently measured in the low nM range [25,26]. In contrast, some spike mutations, such as E484K present in the Beta and Gamma RBDs, abolish binding to certain spike-targeting antibodies [27,28]. Thus, the immune pressures encountered by the virus while transmitting within humans likely underlie the brunt of the observed structural convergence.

The N501Y substitution in the SARS-CoV-2 spike altered interaction with ACE2, observed structurally as well as through infectivity with diverse ACE2 orthologs. It caused sizable shifts in a patch of residues at the ACE2 interaction site, reducing interaction between RBD Gln498 and ACE2 Lys353, while simultaneously enhancing interaction between RBD Gln498 and ACE2 Leu45. It enabled infection of cells harboring an additional quarter of the orthologs within our dataset. This was consistent with the structural analysis of the Alpha RBD, as the distancing of ACE2 Lys353 with the N501Y variant spike is consistent with the K353H difference in mouse ACE2 being a major driver of its incompatibility with Asn501 spike, which can be circumvented with the change to Tyr [29]. The N501Y substitution can epistatically compensate for otherwise detrimental effects caused by other RBD mutations in its absence, as assessed through a deep mutational scanning of the variant spike RBDs [30]. Thus, while N501Y was already known to be critical for SARS-CoV-2 evolution, our results further clarify the manners in which it is important.

Altogether, we speculate a model wherein widespread SARS-CoV-2 replication and evolution in humans enabled shuffling of the molecular determinants underlying compatibility with various animal orthologs of the ACE2 receptor protein (Fig 10). During pandemic spread in humans, SARS-CoV-2 gained mutations in the spike RBD which tweaked its interaction with human ACE2 while simultaneously abrogating binding to common classes of spike-targeting antibodies. These substitutions had outsized effects on altering compatibility for ACE2 orthologs, since many of these interactions were likely of lower affinity and thus straddling the transition point between functionally productive or nonproductive levels of interaction during

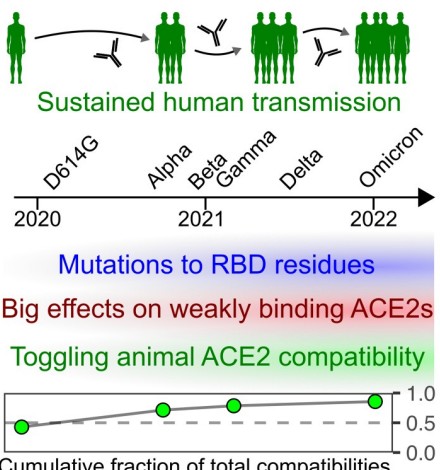

**Fig 10. Speculative model of increased cumulative animal ACE2 compatibility following sustained human transmission.**

viral entry. Since ACE2 ortholog compatibility is a prerequisite for any cross-species transmission event, the amino acid changes caused by immune pressure from one host species, such as prolonged transmission in humans, can promote the virus to tweak its binding surface in ways that coincidentally enable the virus to sample a wider collective range of susceptible animal host species.

## Materials and methods

### Cell culture

All of the cell culture reagents were purchased from ThermoFisher unless otherwise specified. Cell lines were cultured in D10 medium (Dulbecco's modified Eagle's medium supplemented with 10% fetal bovine serum (Gibco,10437028), 100 μg/mL penicillin, and 0.1 mg/mL streptomycin (Corning, 30-002-CI). Cells were passaged via detachment with Trypsin-Ethylenediaminetetraacetic acid 0.25% (Corning, 25-053-CI). Landing pad cells, derived from the LLP-Int-BFP-IRES-iCasp9-Blast construct (Addgene plasmid #171588), was described previously [3]. Landing pad cells were grown in D10 supplemented with 2 μg/mL doxycycline (Fisher Scientific, AAJ67043AD), indicated as D10-dox. For long-term passaging of landing pad cells prior to recombination, the cells were grown in D10-dox with 20 μg/mL blasticidin (InvivoGen, ANT-BL-1) to eliminate cells that silenced their landing pad loci.

### Plasmid construction, barcoding, and pooling

All the barcoded plasmids were produced using Gibson assembly [31]. Many human ACE2 mutant and ACE2 ortholog constructs were created and described in previous publications [3,11], although a handful of additional ACE2 ortholog expression constructs were created for this study using the same molecular cloning strategy previously described. This included incorporation of sequences corresponding to mouse (Addgene # 158087), pig (Addgene # 158085), and pangolin ACE2 (Addgene # 158084). All plasmids were verified to be correct via whole plasmid sequencing provided by Plasmidsaurus, using Oxford Nanopore Technology with custom analysis and annotation. ACE2 ortholog NCBI accession numbers, as well as the protein coding sequences tested in the experiments, are listed in **S3 Table**. For the human

ACE2 mutants expressed at low abundance levels, the "consensus" Kozak / translation initiation sequence of GCCACCATG preceding the protein coding sequence in the high abundance ACE2 ortholog constructs was replaced with the sequence "CATTGTATG", which exhibits substantially reduced translation initiation, which we previously showed to cause a ~50-fold reduced steady-state abundance of the translated protein [3].

For the plasmid barcoding, each clonal plasmid construct was additionally PCR amplified with forward and reverse primers each containing a known 10 nucleotide identifier sequence, together constituting the unique barcode for that construct within the library. Briefly, 30 μL of PCR reactions were created with 40 ng of template DNA, unique primer pair each added to a final concentration of 0.33 μM, and 15 μL of Kapa HiFi HotStart ReadyMix polymerase (Roche Diagnostics, KK2601). Each reaction tube was initially incubated at 95˚C for 5 mins, and then cycled 16 total times with incubations of 98˚C for 20 secs, 65˚C for 15 secs, and 72˚C for 8 mins, followed by a final 72˚C extension of 5 mins. The reactions were incubated with 1μL, or twenty units of DPN1 enzyme (New England BioLabs, R0176L) at 37˚C for 2 hrs. Each reaction was bound and washed on a silica column (Zymo clean and concentrator kit, Zymo Research, D4003) and eluted in water. The barcoded plasmids were circularized by mixing 1μL of the cleaned PCR product with 1μL of Gibson Assembly MasterMix (ThermoFisher, A46629) and incubating the reactions for 60 mins at 50˚C.

The full volume of the resulting products were transformed into E. coli 10β cells (New England BioLabs, C3019I). Following heat-shock, the cultures were spread onto LB-agar plates containing 150 μg/mL of ampicillin, and grown overnight at 37˚C. All of the colonies on the plate were scraped and used to inoculate a 40 mL culture of Luria Broth containing 150 μg/mL of ampicillin (LB-ampicillin), and grown overnight at 37˚C. A subset of the saturated culture was used to extract purified plasmid DNA using a GeneJET Plasmid Miniprep Kit (Thermo Fisher Scientific, K0503).

The efficiency of barcoding was assessed in three different ways. For a subset of samples, at least eight individual colonies were picked from each LB-agar plate prior to scraping, grown in LB-ampicillin, extracted for plasmid DNA, and analyzed with Sanger sequencing (Applied Biosystems 3730 Genetic Analyzer) for the successful barcoding for that clone. For another subset of samples, Sanger sequencing was performed on the scraped plasmid prep, and the chromatogram peak heights for the barcode sequence nucleotides were compared to the peak heights of the nucleotides that would be present in unbarcoded plasmid. For any remaining samples, or previous samples that were ambiguous, nanopore sequencing via Plasmidsaurus was performed on the scraped plasmid mixture, and the raw reads were analyzed to calculate the proportion of barcoded reads as compared to unbarcoded reads. Only plasmid DNA preps that were confirmed to be greater than 75% barcoded were used in downstream experiments, and the barcoding process was repeated for any samples that failed this process until they passed.

All unique plasmid preps that were sufficiently barcoded were eventually mixed together to form the final barcoded plasmid library. Plasmid concentrations were assessed by spectrophotometry, diluted to a final concentration of 100 ng/uL. 30 μL of each of the 33 diluted plasmid preps were mixed together to create the final plasmid DNA mixture that was introduced into the landing pad cells.

### Recombination of barcoded plasmids into landing pad cells

One day prior to transfection, $3.6 \times 10^5$ HEK 293T landing pad cells were added per well for six total wells of a 12-well tissue culture plate. The next day, separate transfection reaction tubes were set up for each well, wherein 2.5 μg of pooled plasmid was diluted with Xfect

reaction buffer (Takara Bio, 631318) to a total volume of 50 µL, followed by addition of 0.6 µL of Xfect polymer. Following incubation, the transfection mixture was added to the cells. The next day, the transfection media was replaced with fresh D10-dox media.

Four days after transfection, negative selection was used to remove the unrecombined cells through the addition of 10nM AP1903 (ApexBio, B4168), which activates the iCasp9 inducible caspase in transcriptionally active, unrecombined landing pad cells. As transcriptionally silenced landing pad cells can survive the negative selection step, the pooled mixture of cells were expanded and subsequently maintained in D10-dox containing 1 µg/mL puromycin (InvivoGen, ANTPR).

## Pseudotyped virus infection assay

All pseudotyped virus infection assays were performed with lentiviral vector cores, coated by one of six viral envelope plasmids: Wuhan/WA1 with D614G (pcDNA6B-SARS2_CoV2-S (D614G;CodonOpt-d19)-FLAG), or the Alpha (Addgene # 170451), Beta (Addgene # 170449), Gamma (Addgene # 170450), Delta (Addgene # 172320), or Omicron BA1 (Addgene # 179907) spikes. Coding sequences of each envelope protein are listed in **S3 Table**. To produce the lentiviral particles, 1 million HEK 293T cells in a single well of a 6-well plate were transfected using PEI-Max MW 40,000 (PolySciences, CAS Number: 49553-93-7) mixed with 1 µg of PsPax2 (Addgene # 12260), 1 µg of the lentiviral transfer vector G1088E_pLenti-CMV-mNeonGreen-2A-HygroR (Addgene #216279), and 1 µg of one of six spike proteins tested.

Twelve hours following transfection, the medium was removed, and cells were gently washed with 1X PBS followed by adding 2 mL D10 medium per well. The media from each well containing pseudotyped lentiviral particles were collected every 24 hours, over the next 72 hours. For each pseudotyped virus produced, this media was pooled, clarified by centrifugation at 1,200 rpm for 3 mins, and the clarified supernatants were separated from cell debris by transferring into a new tube. 2 µg/mL doxycycline and 1 µg/mL puromycin was added to these viral supernatants prior to further use.

For each independent infection experiment, 10 mL or 24 mL of the clarified viral supernatant was mixed with $3.5 \times 10^6$ barcoded ACE2 library -recombined cells in a 10 cm plate. Two days following infection, the cells were visualized under the fluorescence microscope for green fluorescence from mNeonGreen, to confirm that the multiplicity of infection for each plate was approximately at or less than 1. For each infection, roughly 10% of cells were collected and pooled to serve as the unselected control sample, centrifuged at 300 x g for 3 mins, and stored as frozen pellets once the supernatant was removed. The remaining 90% of cells were passaged in D10-dox with 50 µg/mL hygromycin for at least a week. Selection was confirmed by visual inspection that nearly all remaining cells in the dish exhibited green fluorescence, and the cells were collected once the plate reached at least 25% confluence, centrifuged at 300 x g for 3 mins, and stored as frozen pellets once the supernatant was removed. The genomic DNA was extracted from thawed pellets using a DNeasy Blood & Tissue Kit (Qiagen, 69506) or JetFlex Genomic DNA Purification Kit (A30700, Thermo Scientific).

## Amplicon generation and high throughput DNA sequencing

The extracted genomic DNA was used as the template material for PCR amplification of a 490 bp segment containing the plasmid barcode. To avoid sequencing any lingering unintegrated plasmid DNA, a forward primer directly 5-prime of the plasmid barcode was paired with a reverse primer hybridizing to the Bxb1 integrase coding region encoded in the landing pad, so that only integrated barcodes would be amplified. Each sample was amplified with a single 50 µL reaction containing 2 µg of genomic DNA (as assessed by spectrophotometry at $A_{260}$),

0.25uM forward primer containing an i7 index and p7 cluster generator sequence, 0.25uM reverse primer containing i5 index barcode and p5 cluster generator sequence, and 25 μL of Phusion plus high-fidelity PCR mix (Thermo Scientific, F631S). The amplicons were created with an initial 95˚C incubation for 3 mins, and 28 total cycles of 95˚C for 15 secs, 60˚C for 15 secs, 72˚C for 30 secs, followed by a final extension step of 72˚C for 1 min.

The PCR reaction contents were mixed with 6x DNA loading dye (New England Biolabs, B7024S) and separated on 1% agarose gel prepared in Tris-acetate-EDTA buffer, and electrophoresed. Double stranded DNA bands migrating at 490 nt were observed with SYBR Safe DNA Gel Stain (Thermo Scientific, S33102) were excised and extracted with a Freeze and Squeeze kit (Bio-Rad, 7326166) and further purified using a silica column (Zymo Research, D4003). The concentrations of the final DNA were quantified with a Qubit dsDNA BR assay kit (Life technologies, Q32853). Amplicons from all the experimental conditions were pooled together in equimolar ratios, subject to a final quality control step for homogeneity with an Agilent Fragment Analyzer and cluster estimation with quantitative PCR. The DNA was sequenced with a NextSeq 550 using a NextSeq 550 High output 75 cycle or Mid output 150 cycle kit (Illumina).

Following sequencing, the reads were converted to fastq format and de-multiplexed with bcl2fastq. To process the demultiplexed files, we created a custom Python script that extracted the first 10 nucleotides and quality scores from the read1 fastq file, the reverse complement of the first 10 nucleotides and quality scores from the read2 fastq file, and only retained reads that had sequences that matched the known sequences used in the barcoding procedure. Reads that passed this step were output into a single fastq file where the two sets of 10 nucleotides were concatenated as a 20-nucleotide barcode. Files i0192 through i0220, the sixth nucleotide of the read 1 sequences globally exhibited low quality, with many sequences returned with an ambiguous N nucleotide at that position, so the Python script and downstream analyses were modified to retrieve sample identities using a 9 nucleotide character string, and a 19-nucleotide barcode, for those samples.

All resulting fastq files were processed with Enrich2 [32], which was used to tally the counts of all barcodes where the minimum nucleotide quality score was 20 or higher. This process was repeated for every sample, and the resulting barcode counts were written as individual tab-separated value barcode count files for further analysis. The raw Illumina sequencing files, prior to the aforementioned data processing, can be accessed at the NCBI Gene Expression Omnibus (GEO) repository under accession GSE255644.

## Immunoblotting of SARS-CoV-2 variant spike proteins on pseudotyped virus particles and cell lysates

For the immunoblotting experiments, the pseudotyped lentiviral particles were produced as described in the infection assay methods subsection, except that only two separate wells of a six well plate were transfected for each variant spike sample, for each replicate experiment. To capture the pseudotyped virus particles, cell media was collected at 2, 3, and 4 days following transfection, and stored at 4˚C until all samples were collected. To capture the producer cell lysates, upon performing the last media collection, the remaining cells were removed by physical agitation, washed with PBS, and resuspended in 500 μL RIPA buffer (Thermo Scientific, #89901) containing protease inhibitor cocktail (Thermo Scientific, #1862209).

For preparing the pseudotyped virus particle lysates, the 12 mL of stored media collected per sample were centrifuged at $300 \times g$ for 5 minutes, and the clarified supernatants were transferred to a new 15 mL conical tube. One volume of Lenti-X concentrator (Clontech, # 631231) was combined with 3 volumes of clarified supernatant and mixed with gentle

inversion. Mixtures were incubated at 4˚C overnight followed by centrifugation at $1,500 \times g$ for 45 minutes at 4˚C. Supernatants were discarded and the remaining off-white pellets were resuspended in 500 μL of RIPA buffer supplemented with 1X protease inhibitor cocktail.

The cells resuspended in RIPA buffer were incubated for 60 minutes at 4˚C, and then centrifuged at 12,000 rpm for 30 minutes at 4˚C. The clarified lysates were transferred to a fresh tube and quantitated for protein concentration using a BCA assay using a dilution series of bovine serum albumin to create a standard curve (Thermo Scientific, #23223). For the cell lysate, approximately 75 μg of lysate was mixed with 4X SDS sample buffer and boiled for 10 minutes. For the viral particles, equal volumes of lysates were instead used across samples.

All lysates were fractionated with a 4% to 12% gradient SDS PAGE polyacrylamide gel (Genscript, #M00653) followed by transfer to a 0.2-μm PVDF membrane (Thermo Scientific, #88520). Following blocking with 5% dried milk powder resuspended in TBST, the membranes were immunoblotted using antibody against SARS-CoV-2 Spike Protein (S2) (E7L4B, mouse mAb #98603, Cell Signaling, 1:2000 dilution), anti-p24 antibody (Abcam, #63917, 1:2000 dilution) and beta-actin (sc-47778, 1:2000 dilution) antibody. Western blot images were acquired using a Bio-Rad ChemiDoc MP imager. Two independent replicates of this experiment were performed, and the results are shown as **S7 Fig**)

## Data analysis

Subsequent analysis was performed using version 2023.06.0+421 of RStudio running R version 4.2.2. After importing the barcode counts, the relative frequencies of each barcode was calculated by dividing each barcode count by the total count for that sample. If there were two separate amplifications sequenced for the same genomic DNA sample, then the geometric mean frequencies were calculated. The frequency of the barcode in the hygromycin selected sample was divided by the frequency of that same barcode amplified from genomic DNA from cells collected prior to selection, to yield an enrichment score. Each barcode was linked to the construct label at this point.

While each selected sample had its own paired unselected sample, this was largely unnecessary, as there was high correlation in ACE2 cDNA frequencies between samples derived from the same starting culture of library cells (**S6 Fig**), even upon passaging over weeks as multiple infection experiments were being performed. In contrast, there was only moderate correlation between the ACE2 cDNA frequencies across separate library cultures, showing that unselected cells should still be collected from each separate round of infection experiments (**S6 Fig**).

There were nine replicate infections and selection for D614G spike, five replicates for Alpha, two replicates for Beta, two replicates for Gamma, five replicates for Delta, and four replicates for Omicron BA1. The geometric mean was calculated for each set of replicate enrichment scores to yield the final variant-specific enrichment score used for subsequent biological analysis. As indicated in the text and figure legends, subsequent normalization procedures to control constructs encoding human ACE2, or ectodomain-deleted ACE2, both in the absence of TMPRSS2 co-expression, were performed with various analyses to improve analysis interpretability.

An R Markdown file containing code recreating the analysis, the relevant data files necessary to run the code, and the Python scripts needed to process various pieces of data prior to the analysis, can all be found at the Matreyek Lab GitHub repository under "SARS2-CoV-2_ACE2_variant_combinations". Fastq data files can be found at the NCBI GEO repository.

## Structural analysis

We created a Python script which parsed each pdb or cif file and returned atomic coordinates as rows of a tab separated values datafile. These atomic coordinates were imported into R, where chain identifiers were manually entered to filter the atomic coordinates to only one ACE2 polypeptide chain and the singular spike RBD polypeptide chain interacting with it. Using a loop function sequentially testing every residue of ACE2 against every residue of the RBD, the Euclidean distances between every side-chain atom of the ACE2 residue was calculated for every side-chain atom of the RBD residue. The shortest distance value for each pair of residues was recorded into a distance matrix, wherein each row was an ACE2 residue, and each column an RBD residue. The code recreating this aspect of the analysis can be found in the main R markdown file at the Matreyek Lab GitHub repository under "SARS2-CoV-2_ACE2_variant_combinations".

## Supporting information

**S1 Fig. Comparing SARS-CoV-2 D614G pseudotyped virus infectivity with the multiplex assay as compared to more traditional assay formats.** A) Scatter plot showing SARS-CoV-2 D614G spike pseudotyped virus infectivities in cells expressing WT human ACE2, or various human ACE2 mutants, tested through a traditional arrayed infection assay (x-axis) or the multiplex infection assay (y-axis) performed in this work. Infection values were normalized to those from cells expressing WT human ACE2. B) Scatter plot showing SARS-CoV-2 D614G spike pseudotyped virus infectivities in cells expressing human ACE2, or various animal ACE2 orthologs, tested through a traditional arrayed infection assay (x-axis) or the multiplex infection assay (y-axis) performed in this work. Infection values were normalized to those from cells expressing human ACE2. C). Ratios of indicated human ACE2 mutant enrichment scores divided by WT human ACE2 enrichment score, with each replicate value shown for the D614G spike (cyan dot) or Omicron BA1 spike (red dot) coated virus. The mean value across all replicates for each condition is shown as a colored, horizontal line. D). Log-10 transformed enrichment ratios for human and animal ACE2 orthologs expressed at high steady-state abundance, for replicate experiments with D614G spike pseudotyped virus. Individual replicate values are shown as gray points, while geometric mean value across all replicates are shown as horizontal lines.
(TIF)

**S2 Fig. Example plots demonstrating the atomic distances between key ACE2 residues and various residues on SARS-CoV-2 spike RBD D614G.** A) Scatter plots showing the minimal atomic distance (y-axis) between various SARS-CoV-2 spike RBD D614G residues (x-axis) and human ACE2 residue Lys31 (left), Asp38 (middle), or and overlay of the two for comparison (right). B) Line graphs comparing how the minimal distances for ACE2 Lys31 (left) and Asp38 (right) change in regards to each position on the SARS-CoV-2 spike RBD when ensemble structures from various SARS-CoV-2 variants are used. C) Histogram showing the most variable ACE2:RBD minimum distance pairs, as determined by plotting the pairs with the highest coefficients of variation across SARS-CoV-2 RBD variants. The pairs with the highest values are labeled.
(TIF)

**S3 Fig. Selection scheme for identifying key ACE2 and spike RBD residue pairs with structural analysis.** A) Scatterplots comparing atomic distances for each ACE2:RBD residue pair (left), or histograms showing the distribution of difference values that result when the atomic distance of each ACE2:RBD residue pair for one structure was subtracted by the atomic

distance of that same pair in a comparative structure (right). Three example pairs of plots comparing atomic distances between D614G spike RBD residues and ACE2 residues, with the same metrics calculated for Beta, Delta, and Omicron BA1 spikes (top). As all 15 pairwise combinations of SARS-CoV-2 variant spike RBD ensemble structures were considered, we have also shown an example pair of plots demonstrating a comparison between Beta and Omicron BA1 spike RBDs (bottom). B) Histogram showing the combined Z-scores of changes in pairwise residue atomic distances for all combinations of SARS-CoV-2 spike RBD ensemble structures. Histogram bars in red denote residue pairs that exhibited values outside of the central 95% interval of the distribution. C) Venn diagram demonstrating how the 38 key residue pairs were chosen for subsequent structural and functional analysis.
(TIF)

**S4 Fig. Principal component analysis of all ACE2-RBD pairwise distances for all structures analyzed in this work.** Scatter plot with the x-axis displaying the first principal component of the ACE2-RBD residue pairwise distance matrix, and the y-axis showing the second principal component. Shapes of points denote method of acquisition, while the color of the point denotes which RBD variant was solved. PDB identifiers are indicated next to each point.
(TIF)

**S5 Fig. Amino acid comparisons between various *R. sinicus* ACE2 alleles.** A) Protein sequence comparisons showing residue differences between various chinese horseshoe bat ACE2 alleles, with major differences from the most common amino acid highlighted in red. The human amino acid at each position is shown for comparison, with major differences highlighted in blue. Residues located in the signal peptide or in the first alpha helix of ACE2 are labeled above the curated alignment. B) Tertiary structure of the ACE2 surface interacting with the SARS-CoV-2 spike, highlighting residues that are variable within the *R.sinicus* alleles. PDB entry 6m17, consisting of a cryo-EM structure of human ACE2 bound with the SARS-CoV-2 spike RBD, was used to show the three-dimensional locations of these highly variable residues. The full ACE2 polypeptide is shown as a cartoon representation, with the amino acid side chains for the human ACE2 protein are shown as spheres. C) Same tertiary structure representation as in panel B, but with the SARS-CoV-2 spike RBD shown as a cyan cartoon representation, including key interaction residues highlighted in our structural analysis shown as spheres.
(TIF)

**S6 Fig. Unselected recombined barcode correlations following prolonged passaging or independent freeze-thaws.** Set 1 consists of a group of 17 different subpopulations collected as "unselected cells" over the course of two weeks of experiments, derived from the same vial of frozen, recombined cells that were thawed at the start of that series of experiments. Set 2 includes samples derived from an independent thaw of the same library of cells several months later. The 16 "unselected cell" samples were collected over the course of two weeks, spanning December 22, 2022 through January 4, 2023. The histograms show the Pearson's r-squared values resulting from all pairwise comparisons of all unselected barcode counts sequenced from set 1 or set 2 genomic DNAs, but displayed as different histograms based on whether the correlations were calculated for two samples within the same set or within separate sets.
(TIF)

**S7 Fig. Immunoblots showing SARS-CoV-2 spike variant expression and incorporation into pseudotyped viral particles.** The paired producer cell lysate and viral particle immunoblotting experiments were performed in two independent replicates. Cell lysate was immunoblotted with a SARS-CoV-2 spike S2-fragment targeting antibody and a beta-actin loading

control (top), whereas pelleted pseudotyped virus particles were blotted with the same spike antibody and an antibody targeting the lentivector capsid core (p24) as the viral particle loading control (bottom).
(TIF)

**S1 Table. Information for each construct and the primers used to create the barcode.**
(CSV)

**S2 Table. PDB entry identifiers and acquisition types used in the structural analysis.**
(CSV)

**S3 Table. Accession numbers and protein sequences used in this study.**
(CSV)

**S4 Table. Sequencing file names and the sample replicate they corresponded to.**
(CSV)

**S5 Table. Primer names and the partial barcode sequences they encode.**
(CSV)

## Acknowledgments

We wish to thank Simone Edelheit and Milena Zelembaba of the Genomics Core Facility of the CWRU School of Medicine's Genetics and Genome Sciences Department. This research was supported by the Cytometry & Imaging Microscopy Shared Resource of the Case Comprehensive Cancer Center (P30CA043703 and S10OD021559).

## Author Contributions

**Conceptualization:** Kenneth A. Matreyek.

**Data curation:** Kenneth A. Matreyek.

**Formal analysis:** Kenneth A. Matreyek.

**Funding acquisition:** Kenneth A. Matreyek.

**Investigation:** Nidhi Shukla, Sarah M. Roelle, Olivia DelSignore, Kenneth A. Matreyek.

**Methodology:** Nidhi Shukla, John C. Snell, Kenneth A. Matreyek.

**Project administration:** Kenneth A. Matreyek.

**Resources:** Kenneth A. Matreyek.

**Software:** Kenneth A. Matreyek.

**Supervision:** Kenneth A. Matreyek.

**Validation:** Kenneth A. Matreyek.

**Visualization:** Kenneth A. Matreyek.

**Writing – original draft:** Kenneth A. Matreyek.

**Writing – review & editing:** Nidhi Shukla, Sarah M. Roelle, John C. Snell, Anna M. Bruchez, Kenneth A. Matreyek.

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
