## [Decision Letter · Decision Letter 0]

29 Feb 2024

Dear Dr. Matreyek,

Thank you very much for submitting your manuscript "Pseudotyped virus infection of multiplexed ACE2 libraries reveals SARS-CoV-2 variant shifts in receptor usage" for consideration at PLOS Pathogens. As with all papers reviewed by the journal, your manuscript was reviewed by members of the editorial board and by several independent reviewers. The reviewers appreciated the attention to an important topic. Based on the reviews, we are likely to accept this manuscript for publication, providing that you modify the manuscript according to the review recommendations.

Sincerely,

Lijun Rong, Ph.D

Guest Editor

PLOS Pathogens

Debra Bessen

Section Editor

PLOS Pathogens

Michael Malim

Editor-in-Chief

PLOS Pathogens

orcid.org/0000-0002-7699-2064

Reviewer Comments (if any, and for reference):

Reviewer's Responses to Questions

**Part I - Summary**

Reviewer #1: Shukla et al. describes a novel high-throughput assay to simultaneously study the interactions between the SARS-CoV-2 spike protein with multiple ACE2 proteins. A library of 293T cells was created with each cell expressing one of 30 different DNA-barcoded ACE2 orthologs. These cells were subsequently infected with a SARS-CoV-2 spike-pseudotyped virus carrying a hygromycin resistant gene, followed by hygromycin selection and Illumina sequencing to identify enriched ACE2 orthologs. A total of 6 SARS-CoV-2 variants were tested, generating a comprehensive dataset that was used to characterize the structure and function relationship between SARS-CoV-2 variants and ACE2 orthologs. The data produced with this multiplex assay were largely in agreement with published data produced with the traditional assays.

The development of a multiplex assay to study the compatibility of SARS-CoV-2 spike proteins and ACE2 orthologs in large scale is innovative and can have a broad usage for other viruses. The paper is well written and the results are clearly presented. However, several issues need to be addressed.

Reviewer #2: This study describes an innovative experimental system to investigate viral glycoprotein interactions with their cellular receptors, using a lentivirus pseudotyping system and barcoded transgenic receptor constructs. The power of the system is demonstrated by example of SARS-CoV-2 spike representing the original isolate and a selection of VOC and human ACE2 and 13 animal ACE2 orthologs. Major strengths are the demonstrated prediction of shifts in spike variant interactions with human ACE2 and of spike compatibility with animal ACE2 orthologs.

Results are clearly presented, although the Results section of the manuscript would benefit from greater focus on the main findings. Overall, this study advances the understanding of the combinatorial sequence space between SARS-CoV-2 spike and ACE2, and introduces an exciting experimental approach with major implications extending broadly to glycoprotein evolution and receptor tropism of enveloped viruses.

**Part II – Major Issues: Key Experiments Required for Acceptance**

Reviewer #1: 1. The expression level of each spike protein needs to be examined.

2. Fig. 2A: the fold enrichment in the human ACE2 orthologs group is very small. How many independent experiments were performed? Were the differences statistically significant?

3. Fig. 5D: there seem to be big variations between the two repeats for human ACE2. Were the results for other ACE2 orthologs reproducible?

Reviewer #2: The application of the strategy to predict spike compatibility with animal ACE2 orthologs is intriguing, and the use of a large number of orthologs in the study appreciated. The rationale for ortholog selection is unclear, however, and needs to be better defined. Of great interest would have been compatibility predictions for animal species in close contact with humans and species used frequently as animal models. If additional ACE2 orthologs were examined, inclusion of these data would substantially increase the impact of the manuscript.

**Part III – Minor Issues: Editorial and Data Presentation Modifications**

Reviewer #1: 1. Fig. 2A: please explain how the different level of ACE2 protein expression was achieved.

2. Lines 390-392: these sentences are misleading. Natural infections occur with mixed populations of cells, but that doesn’t mean the receptors are different.

3. Fig. 10 and last paragraph in Discussion: the model that sustained transmission among humans leads to increased animal ACE2 compatibility seems to be highly speculative.

Reviewer #2: N/A

PLOS authors have the option to publish the peer review history of their article (what does this mean?). If published, this will include your full peer review and any attached files.

Reviewer #1: No

Reviewer #2: No

Figure Files:

Data Requirements:

Reproducibility:

References:

---

## [Editor Report · Decision Letter 1]

7 May 2024

Dear Dr. Matreyek,

We are pleased to inform you that your manuscript 'Pseudotyped virus infection of multiplexed ACE2 libraries reveals SARS-CoV-2 variant shifts in receptor usage' has been provisionally accepted for publication in PLOS Pathogens.

Best regards,

Lijun Rong, Ph.D

Guest Editor

PLOS Pathogens

Debra Bessen

Section Editor

PLOS Pathogens

Michael Malim

Editor-in-Chief

PLOS Pathogens

orcid.org/0000-0002-7699-2064
---

## [Editor Report · Acceptance letter]

13 May 2024

Dear Dr. Matreyek,

We are delighted to inform you that your manuscript, "Pseudotyped virus infection of multiplexed ACE2 libraries reveals SARS-CoV-2 variant shifts in receptor usage," has been formally accepted for publication in PLOS Pathogens.

Best regards,

Michael Malim

Editor-in-Chief

PLOS Pathogens

orcid.org/0000-0002-7699-2064